# Implementation of cervical cancer prevention and screening across five tertiary hospitals in Nepal and its policy implications: A mixed-methods study

Ganesh Dangal[1,2], Rolina Dhital[3]*, Yam Prasad Dwa[1,4‡], Sandesh Poudel[1,5‡], Jitendra Pariyar[1,6‡], Kirtipal Subedi[1,5‡]

1 Nepal Society of Obstetricians and Gynaecologists, Kathmandu, Nepal, 2 Kathmandu Model Hospital, Kathmandu, Nepal, 3 Health Action and Research, Kathmandu, Nepal, 4 KIST Medical College and Teaching Hospital, Lalitpur, Nepal, 5 Paropakar Maternity and Women's Hospital, Kathmandu, Nepal, 6 Civil Service Hospital of Nepal, Kathmandu, Nepal

☯ These authors contributed equally to this work.
‡ YPD, SP, JP and KS also contributed equally to this work.
* rolina.dhital@gmail.com

**Data Availability Statement:** We have uploaded the quantitative data to Figshare with a private link for Editorial and Review Purpose. The link to

## Abstract

In Nepal, cervical cancer is the most common cancer among women despite the existing policies. This study intends to assess the implementation of cervical cancer prevention and screening through service utilization by women, knowledge and attitude among health professionals, and the perceptions of stakeholders in Nepal. This mixed-methods study was conducted in 2022 across five tertiary hospitals in Kathmandu, Nepal. The quantitative study comprised the health professionals and women attending gynecology outpatient clinics from the selected hospitals. The qualitative study comprised stakeholders including service providers and experts on cervical cancer from selected hospitals, civil societies, and the Ministry of Health and Population. The utilization of screening through pap smear among 657 women across five hospitals was 22.2% and HPV vaccination was 1.5%. The utilization of cervical cancer screening was associated with older age [adjusted odds ratio (AOR) = 1.09, CI: 1.07, 8.19], married (AOR = 3.024, CI: 1.12, 8.19), higher education (AOR = 3.024, CI:1.12, 8.42), oral contraceptives use (AOR = 2.49, CI: 1.36, 4.39), and ever heard of cervical cancer screening (AOR = 13.28, CI: 6.85, 25.73). Among 254 health professionals, the knowledge score was positively associated with them ever having a training [Standardized Beta (β) = 0.20, CI: 0.44, 2.43)] and having outreach activities in their hospital (β = 0.19 CI: 0.89, 9.53) regarding cervical cancer screening. The female as compared to male health professionals (β = 0.16, CI: 0.41, 8.16, P = 0.03) and having a cervical cancer screening guideline as compared to none (β = 0.19 CI: 0.89, 9.53, P = 0.026) were more likely to have a better attitude for screening. The qualitative findings among 23 stakeholders reflected implementation challenges in policy, supply, service delivery, providers, and community. This study showed low utilization of prevention and services by women and implementation gaps on cervical cancer prevention and screening services across five tertiary hospitals in Kathmandu, Nepal. The findings could help designing more focused interventions.

dataset is https://figshare.com/s/e9247100ffb9900b7099. We shall make the link public upon acceptance. For qualitative data, underlying data is available within the manuscript itself and in supplementary files.

**Funding:** The authors received no specific funding for this work.

**Competing interests:** The authors have declared that no competing interests exist.

## Introduction

Globally, cervical cancer is the fourth most common cancer in women and continues to be a global public health concern [1]. In 2020, there were an estimated 604,000 new cases of cervical cancer worldwide and about 342,000 women died from the disease [1]. When diagnosed early, cervical cancer is one of the most successfully treatable forms of cancer, as long as it is detected early and managed effectively [2]. The common interventions for cervical cancer screening and prevention include community-based awareness activities and early medical interventions [3–7]. The community-based awareness activities include home visits for raising awareness and cultural awareness by health professionals, educational interventions, and media campaigns about prevention of cervical cancer [3–6]. The common early medical interventions can be grouped as screening, Human Papilloma Virus (HPV) vaccination, and treatment [8]. The screening includes pap smear, visual inspection with acetic acid (VIA), and HPV DNA testing [8].

The systematic reviews on cervical cancer have indicated screening of cervical cancer to be a cost-effective intervention globally in timely diagnosis and treatment even in resource limited settings [4–7, 9–13]. Studies globally have also identified barriers to cervical cancer screening which include lack of education, low socioeconomic status, lack of knowledge, lack of effective communication, embarrassment, time constraints, and preference for female doctors [14–16]. The studies also highlighted under-representation across different countries, mostly in poorer countries within each region of Africa, Europe, and Asia [5, 11, 13]. Moreover, the sustainability of the interventions remained inconclusive in the long run.

In Nepal, cervical cancer is the most common cancer among women, accounting for the highest mortality rate (9.46 per 100,000 population) among all HPV related cancers in Nepal [17]. The age-standardized yearly incidence of cervical cancer in Nepal is 16.4 per 100,000, making it one of the countries in South Asia with the highest cervical cancer rates followed by India and Bangladesh [17]. The national guidelines on cervical cancer screening and prevention were introduced in 2010 in Nepal. It suggested screening at least 50% of women aged 30–60 years, with recommended screening every five years to reduce cervical cancer mortality by 10% [18].

Most of the studies conducted in Nepal are single-centered hospital-based studies or community-based observational studies [19–25]. The knowledge and utilization of cervical cancer screening remained low in most studies [20–25]. The identified barriers aligned with the global literature and included poor literacy, lack of awareness, sociocultural aspects, and embarrassment [14–16, 19–25]. Studies recommended culturally contextual and tailored interventions to improve knowledge and practices [19–25]. However, a comprehensive multicenter study reflecting the users' and providers' perspectives remains scant.

This study intended to assess the implementation of cervical cancer prevention and screening services in five tertiary hospitals in Kathmandu, Nepal. The implementation is assessed through three perspectives: the utilization of prevention and screening services by women, the knowledge and attitude of service providers regarding cervical cancer prevention services and screening, and the perceptions of stakeholders working on cervical cancer. The findings could help generate baseline evidence at hospital levels, identify specific areas to improve and outline the way forward for sustainable interventions.

## Methods

### Study design

This is a mixed-methods research design that followed a sequential explanatory method [26]. The study followed two distinct phases where the first phase comprised quantitative data

collection and analysis, and the second phase comprised the qualitative data collection and analysis [27]. The qualitative approach intended to provide deeper insights into the quantitative findings obtained in the first phase [26, 27]. The integration of the quantitative and qualitative findings was done at the interpretation level in the discussion [26, 27]. Fig 1 summarizes the study flow of the sequential explanatory method (Fig 1).

## Study setting

This study was conducted in five tertiary hospitals in Kathmandu, Nepal representing government, non-government and private, and teaching hospitals. The hospitals were selected purposively based on the patient flow, feasibility, and accessibility for conducting research. The five hospitals are Paropakar Maternity and Women's Hospital (PMWH), Civil Service Hospital of Nepal (CH), KIST Medical College Teaching Hospital (KIST), Kathmandu Model Hospital (KMH), and Kirtipur Hospital (KH). We selected only the tertiary hospitals in this study as most tertiary care centers provide cervical cancer screening services and have a high patient flow.

The screening of cervical cancer in Nepal is provided at both population level as well as at hospital level. The population-based screening is done through outreach health camps in the communities mostly organized by the hospitals in collaborations with the government and non-government organizations [28]. In the hospital setting, most hospitals have the screening service as part of the regular outpatient clinic services. Pap smear test remains the most commonly used method for both population-based and hospital-based cervical cancer screening in Nepal [28]. The screening with VIA and HPV DNA tests have not been able to gain wide coverage resulting in low service coverage in Nepal with only specialized higher centers providing such screening services [28]. Therefore, we chose utilization of screening services through pap smear in the selected hospitals as it is the most commonly available screening services in Nepal and also provided by all the selected hospitals in this study. The screening across all the selected hospitals were performed by the medical doctors and assisted by the nurses. The treatment of pre-cancerous lesions is provided by all the selected hospitals whereas treatment of the advanced cancer is often referred to the specialized cancer hospitals by the doctors.

## Study participants

For the quantitative study, the participants comprised the health professionals and women attending gynecology outpatient clinics of the selected hospitals. The health professionals included the medical doctors and nurses currently working in the obstetrics and gynecology departments of the selected hospitals.

For the qualitative study, the participants comprised the stakeholders representing each of the selected hospitals, representatives from gynecology and oncology societies, and the Ministry of Health and Population (MOHP) and Department of Health Services who are responsible or may have the influence in bringing policy changes related to cervical cancer prevention services.

## Sample size

**Quantitative study.** For health professionals, the minimum required sample size was calculated based on the knowledge score from a cross-sectional study that was conducted in Uganda [29] as we had adapted the study tools on knowledge and attitude from the same study. The minimum required sample was calculated to be 241 with a margin error of 5% and a confidence interval (CI) of 95%. Considering a 10% non-response rate, the required sample size was calculated to be 268.

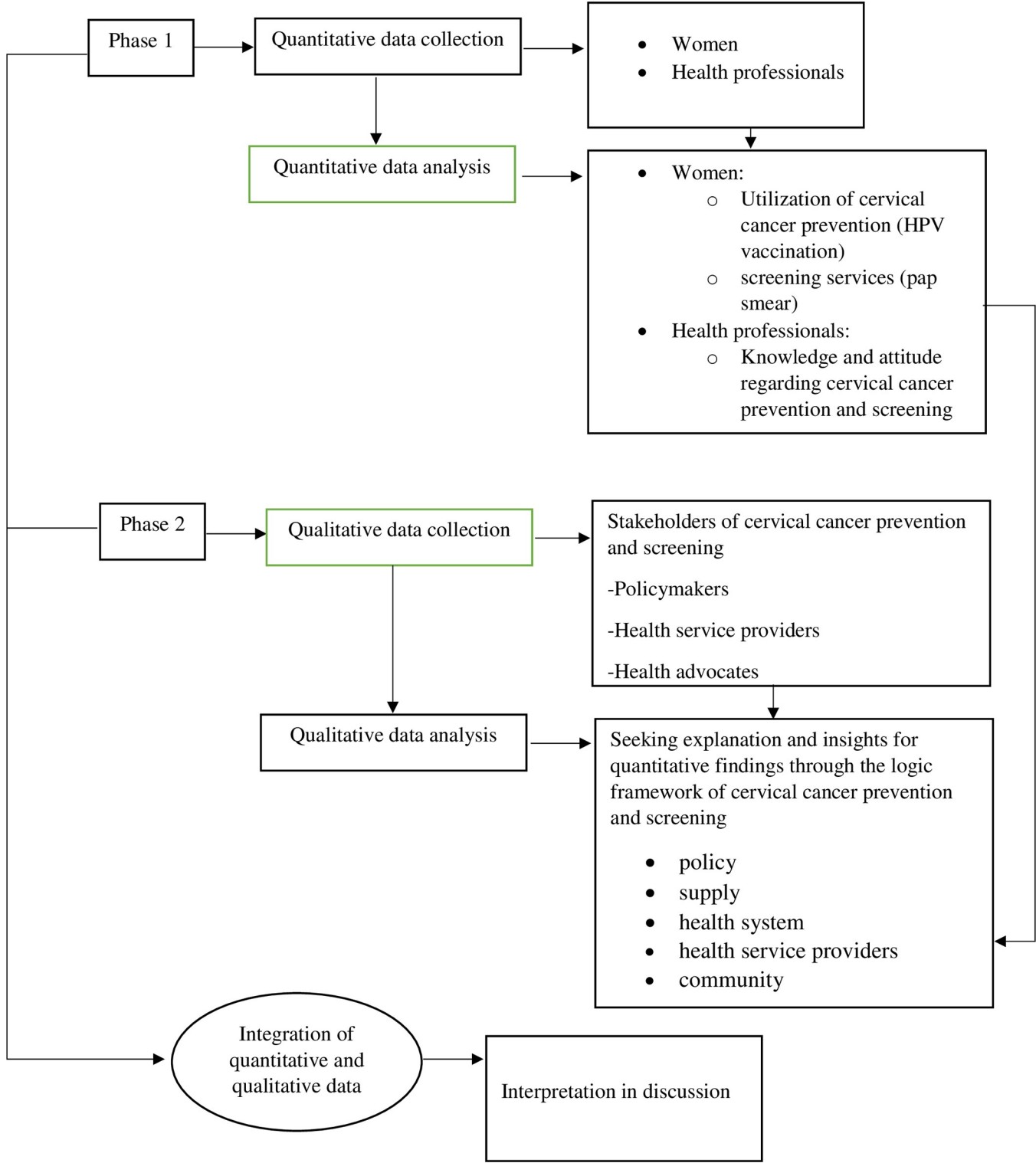

**Fig 1. Study flow of the sequential explanatory method.**

For the women attending gynecology outpatient clinic, we calculated the sample size based on a previous study conducted in Nepal that had reported proportion of 44.9% women utilizing of cervical cancer screening [21]. The minimum required sample was calculated to be 379 to report the utilization of cervical cancer screening, with a margin of error of 5% and a CI of 95%. Considering that this study will be a multicenter study, we consider each hospital as a cluster and increase the sample size by considering design effects, with a within-hospital-intraclass correlation coefficient of 0.01 for a multicenter study [30]. We considered each hospital as a cluster, and the average number of patients attending gynecology outpatient clinic per day was estimated to be 40. Then we calculated the design effect as 1.39, using the formula Deff = 1 +(m-1)×p, where m represents the average number of patients attending gynecology outpatient clinic each day, and p is the intra-class correlation coefficient. After multiplying the minimum sample size by the design effect, the required number of participants was 526. However, considering a 20% non-response rate, the sample was increased to 657.

**Qualitative study.**    The data was collected until the information reached the point of saturation. In total, data were collected from 23 participants.

## Sampling

**Quantitative study.**    For health workers, data was collected through a stratified random sampling method. The health workers were stratified into the doctors' and nurses' groups. A sampling frame was created for each stratum in each hospital based on the complete list of all eligible medical doctors and nurses currently working in the obstetrics and gynecology department of each hospital. The eligible doctors and nurses in each hospital were identified by the authors (GD, YPD, SP, JP, and KS) representing each hospital based on the administrative records of the hospitals. A unique number code starting from '1' in consecutive orders was given to the list of doctors and nurses separately in each hospital in an excel sheet for each hospital. A separate anonymized file with the coded numbers for each stratum was created for each hospital. From the sampling frame of participants, 60% of the eligible participants for each stratum in each hospital were selected randomly through computer-generated random numbers by an independent researcher (RD) not representing any hospital. The doctors and nurses from the selected random number codes for each stratum in each hospital were invited to participate in the study by the research assistants in each hospital. In case of refusal to participate, a new random number was generated to invite a new participant for the replacement of the ones who refused to participate.

For women, data was collected through convenient sampling as it was not feasible to create a sample frame for random sampling as the selected hospitals had high patient flow, no prior appointment were needed for patients to attend the outpatient clinic, and employing the random sampling techniques would have been difficult given the time and resource limitations. Women attending the gynecology outpatient clinic who agreed to participate in the study were interviewed from each hospital. The research assistants in each hospital collected data from 131 to 132 women from the outpatient clinic during a period of two weeks with an average of 9 to 10 participants enrolled in the study each day in each facility.

**Qualitative study.**    For the qualitative study, the purposive sampling method was performed based on a potential list of participants who had an important role in cervical cancer screening at hospitals and policy levels.

## Study variables

The dependent variable for women attending the outpatient clinic were their utilization of HPV vaccination and cervical cancer screening through a pap smear. The independent

variables included sociodemographic characteristics and if they have ever heard of cervical cancer, screening of cervical cancer, and HPV vaccination.

The dependent variables for health professionals included their knowledge of cervical cancer and their attitude toward cervical cancer screening which was assessed by adapting a validated tool used in a study from Uganda [29]. The independent variables included sociodemographic and professional characteristics, training, and the availability of resources for cervical cancer prevention services and screening.

## Operational definitions and measurements

**Utilization of cervical cancer screening.** Utilization of cervical cancer screening was assessed among women attending the gynecology outpatient clinic in the selected hospitals through "Yes" (1) or "No" (0) question on if they had ever utilized the screening service for cervical cancer via pap smear. The question was measured as a dichotomous variable in our study.

**Uptake of HPV vaccination.** The question was asked to the women if they had ever had HPV vaccination. The responses were either "Yes (1)" or "No (0)" and measured as a dichotomous variable.

**Knowledge of cervical cancer among health professionals.** The knowledge of cervical cancer among health professionals was assessed through a scale comprising 25 questions [29] focusing on three subtopics that included knowledge on risk factors for cervical cancer, signs and symptoms of cervical cancer, and cervical cancer prevention (S1 Table). A score of "1" was given to each correct answer and a score of "0" was given to each wrong answer. The scores were summed up with a total possible score of 25 and the mean score was calculated as it was treated as a continuous variable. The higher scores indicated better knowledge.

**Attitude toward cervical cancer screening.** The attitude toward cervical cancer screening was measured through a total of 12 questions that focused on the health professionals' willingness to participate in training, screening and prevention activities (S2 Table). The original tool comprised 13 questions [29] but we removed one question in our study as the question had focused on the training to be provided by the local university in Uganda which was not applicable in Nepal's context. The responses to the questions were in Likert scales with 5 options ranging from "strongly disagree" (1), "disagree" (2), "neutral" (3), "agree" (4), and "strongly agree" (5). The scores for each question were summed up with a total possible score of 60. The mean scores were calculated as it was measured as a continuous variable in our study with higher score indicating better attitude toward cervical cancer screening.

## Data collection

The data was collected for quantitative and qualitative studies between March 4, 2022 and May 26, 2022.

**Quantitative study.** The health professionals filled in the self-administered online questionnaire developed in the Kobo toolbox [31]. The questionnaire was sent to the selected doctors and nurses by the research assistants via emails and direct messages in social media.

The women attending the gynecology outpatient clinic of each hospital were interviewed by the research assistants. The research assistant used a mobile or tablet-based questionnaire using the Kobo toolbox [31]. The research assistants were trained in research ethics and data collection.

**Qualitative study.** The data collection for the qualitative study adhered to the Consolidated criteria for reporting qualitative research (COREQ) guidelines (S1 Checklist) [32]. A qualified female public health graduate trained in conducting key informant interviews (KII)

took the interviews with the stakeholders. The interviews lasted from 30 minutes to one hour. All interviews were audio recorded with the consent of the participants. The interviewer introduced herself and developed a rapport with the participants. At the end of the interviews, the interviewer summarized the key points with the participants for their feedback.

**Data analysis.** The data analysis for this study followed a sequential explanatory method where the preliminary data analysis was first performed for the quantitative data [27]. The qualitative data were collected only after the preliminary results were obtained from the quantitative data. The qualitative data collection and analysis aimed at explaining and providing more insights into the quantitative findings [27]. The integration of the quantitative and qualitative data was done at the interpretation level in the discussion (Fig 1).

**Quantitative analysis.** The quantitative data were analyzed using SPSS version 23 [33]. A descriptive analysis of the general characteristics was performed for both the women and health professionals. For women's data, unadjusted and adjusted logistic regression was performed to assess the factors associated with the utilization of cervical cancer screening through a pap smear. For health professionals' data, linear regression was performed to assess the factors associated with knowledge and attitude regarding cervical cancer screening.

**Qualitative analysis.** Thematic content analysis was performed for qualitative data using Dedoose 8.3.45 [34]. The logic model for comprehensive, inter-sectoral cervical cancer prevention based on a scoping review was used as a theoretical framework [35]. The categories were identified by the second author (RD) from the coding of the transcripts which were then fitted into five major themes of the logic model framework for cervical cancer prevention services. The themes and categories were finalized by all authors. Original anonymous quotes were included to provide more insights.

## Ethics statement

This study was approved by Nepal Health Research Council (Proposal ID 34–2022). The study also obtained approval from the institutional review boards of all five study hospitals. Written informed consent was obtained from the participants and the participation was voluntary. The confidentiality of the participants was maintained.

## Results

### Quantitative results

Table 1 shows the characteristics of the women attending the outpatient clinic of the five hospitals. The mean age of the women was 37.1 (SD 12.2) with the youngest being 18 years and the oldest 79 years of age. The majority of the women were married (81.4%), of Brahmin/Chhetri ethnic groups (50.5%), and were from the Hindu religion (91%). A family history of cancer was reported by 17.8% of women. In total 17.9% were smokers, 22.5% suggested they have used oral contraceptives, and 3.8% mentioned they have multiple sexual partners. Around 16% of women reported that they had ever heard about HPV vaccination with only 1.5% reporting that they had ever been vaccinated against HPV. Over 50% of women reported that they had ever heard about cervical cancer screening with only 22.2% women ever utilizing the screening service through pap smear.

Table 2 shows the characteristics of the health professionals from the five hospitals. A total of 254 health professionals from the five hospitals participated in the study, of which 171 were nurses and 83 were doctors. The mean age of the participants was 31.9 (SD 7.8) with the youngest being 20 years of age and the oldest 58 years of age. The majority of the respondents were female (87.8%).

**Table 1. Characteristics of women.**

| Characteristics | (N = 657) | % |
|---|---|---|
| | n | |
| **Age in years (mean SD)** | 37.1 | 12.2 |
| **Marital status** | | |
| Married | 535 | 81.4 |
| Unmarried | 122 | 18.6 |
| **Education** | | |
| Masters | 59 | 9 |
| Undergraduate | 176 | 26.8 |
| Secondary education | 143 | 21.8 |
| Primary education | 106 | 16.1 |
| Literate | 74 | 11.3 |
| Illiterate | 99 | 15.1 |
| **Ethnicity** | | |
| Brahmin/Chhetri | 332 | 50.5 |
| Janajati | 228 | 34.7 |
| Dalit | 40 | 6.1 |
| Others | 50 | 7.6 |
| **Religion** | | |
| Hindu | 598 | 91.0 |
| Buddhist | 42 | 6.4 |
| Others | 17 | 2.5 |
| **Employment status** | | |
| Employed | 222 | 33.8 |
| Housewife | 309 | 47.0 |
| Agriculture occupation | 50 | 7.6 |
| Daily wage worker | 39 | 5.9 |
| Others | 37 | 5.6 |
| **Family history of cancer** | | |
| Yes | 117 | 17.8 |
| No | 540 | 82.2 |
| **Smoking** | | |
| Yes | 91 | 13.9 |
| No | 566 | 86.1 |
| **Oral Contraceptive Use** | | |
| Yes | 148 | 22.5 |
| No | 509 | 77.5 |
| **Multiple sexual partners** | | |
| Yes | 25 | 3.8 |
| No | 632 | 96.2 |
| **Ever heard of HPV vaccination** | | |
| Yes | 111 | 16.9 |
| No | 546 | 83.1 |
| **Ever vaccinated against HPV** | | |
| Yes | 10 | 1.5 |
| No | 647 | 98.5 |
| **Ever heard of cervical cancer screening** | | |
| Yes | 351 | 53.4 |

(*Continued*)

**Table 1.** (Continued)

| Characteristics | (N = 657) | % |
|---|---|---|
|  | n |  |
| No | 306 | 46.6 |
| **Ever been screened for cervical cancer via pap smear** |  |  |
| Yes | 146 | 22.2 |
| No | 511 | 77.81 |

Only 37% of the health professionals responded that they had ever been formally trained in conducting screening for cervical cancer. In total, 79.1% responded that they have cervical cancer guidelines for screening in their hospital, 88.6% responded that they have education materials related to cervical cancer screening in their hospital, and 88.6% responded that they have outreach activities for cervical cancer screening in their hospitals.

The total mean score for knowledge was 19.8 ± 3.5 out of total possible score of 25. The total mean score for attitude was 47.9 ±8.8 out of total possible score of 60.

Table 3 demonstrates the unadjusted odds ratios (UOR) and adjusted odds ratios (AOR) in logistic regression models. The model was adjusted for the hospitals, sociodemographic characteristics, risk factors, and if they had heard about cervical cancer screening. In the adjusted model, utilization of cervical cancer screening was found to be associated with an increase in age (AOR = 1.13, CI: 1.07, 1.13, P<0.001). Married women were three times more likely to

**Table 2.** Characteristics of the health professionals.

| Characteristics | N = 254 | % |
|---|---|---|
|  | n |  |
| **Age (Mean ± S.D) (Maximum, Minimum)** | 31.9 ± 7.8 | (20, 58) |
| **Gender** |  |  |
| Male | 31 | 12.2 |
| Female | 223 | 87.8 |
| **Occupation** |  |  |
| Nurse | 171 | 67.3 |
| Doctor | 83 | 32.7 |
| **Years of work experience (Mean ± S.D)** | 7.5 ± 7.0 |  |
| **Ever been trained on how to conduct screening for cervical cancer?** |  |  |
| Yes | 94 | 37 |
| No | 160 | 63 |
| **Does the health facility have any guidelines for cervical cancer screening?** |  |  |
| Yes | 201 | 79.1 |
| No | 53 | 20.9 |
| **Does this health facility have health education material about cervical cancer?** |  |  |
| Yes | 225 | 88.6 |
| No | 29 | 11.4 |
| **Do you conduct outreach health education in the community for cervical cancer?** |  |  |
| Yes | 225 | 88.6 |
| No | 29 | 11.4 |
| * Knowledge score out of a total of 25 scores (Mean ± S.D) | 19.8 ± 3.5 |  |
| * Attitude score out of a total of 60 scores (Mean ± S.D) | 47.9 ±8.8 |  |

*Descriptive data for each question is provided in S1 and S2 Tables

**Table 3. Factors associated with utilization of cervical cancer screening among women.**

| | UOR | 95% CI | AOR[a] | 95% CI |
|---|---|---|---|---|
| **Hospitals** | | | | |
| CH | 1.48 | (0.96, 2.29) | 1.21 | (0.51, 2.83) |
| KMH | 2.30 | (1.15, 3.50)*** | 2.34 | (0.97, 5.62) |
| KH | 0.65 | (0.39, 1.07) | 1.36 | (0.54, 3.44) |
| PMWH | 0.89 | (0.55, 1.42) | 1.33 | (0.53, 3.33) |
| KIST | Ref | | Ref | |
| **Age** | 1.05 | (1.04,1.07)*** | 1.09 | (1.07, 1.13)*** |
| **Marital status** | | | | |
| Married | 6.85 | (2.95, 15.92)*** | 3.02 | (1.12, 8.19)* |
| Unmarried | Ref | | | |
| **Education** | | | | |
| Masters | 1.09 | (0.58, 2.06) | 2.73 | (0.79, 9.39) |
| Undergraduate | 0.68 | (0.44, 1.06) | 3.04 | (0.98, 9.36) |
| Secondary education | 1.88 | (1.24, 2.84)** | 5.07 | (1.82, 14.09) |
| Primary education | 1.32 | (0.82, 2.12) | 3.06 | (1.12, 8.42) |
| Literate | 0.79 | (0.43, 1.47) | 1.61 | (0.54, 4.78) |
| Illiterate | Ref | | | |
| **Ethnicity** | | | | |
| Brahmin/Chhetri | 1.29 | (0.89, 1.87) | 1.07 | (0.42, 2.73) |
| Janajati | 1.01 | (0.69, 1.49) | 1.21 | (0.45, 3.27) |
| Dalit | 0.48 | (0.18, 1.25) | 0.66 | (0.15, 2.81) |
| **Occupation** | | | | |
| Employed | 0.99 | (0.67,1.46) | 0.75 | (0.30, 1.84) |
| Housewife | 1.39 | (0.96, 2.01) | 0.97 | (0.41, 2.28) |
| Agriculture | 0.28 | (0.10, 0.80) | 0.35 | (0.08, 1.50) |
| Daily wage workers and others | Ref | | | |
| **Religion** | | | | |
| Hindu | 1.272 | (0.64, 2.52) | 4.02 | (0.37, 43.49) |
| Buddhist | 1.101 | (0.53, 2.29) | 5.09 | (0.40, 64.60) |
| Others | Ref | | | |
| **Family History of Cancer** | | | | |
| Yes | 2.59 | (1.68, 3.99)*** | 1.35 | (0.76, 2.39) |
| No | Ref | | Ref | |
| **Smoking** | | | | |
| Yes | 0.54 | (0.29, 1.01) | 0.86 | (0.36, 2.08) |
| No | Ref | | | |
| **Oral Contraceptive Use** | | | | |
| Yes | 1.927 | (1.28, 2.90)** | 2.44 | (1.36, 4.39)** |
| No | Ref | | | |
| **Multiple sexual partners** | | | | |
| Yes | 1.379 | (0.56, 3.37) | 1.23 | (0.31, 4.84) |
| No | Ref | | | |
| **Ever heard of cervical cancer screening** | | | | |
| Yes | 9.184 | (5.44, 15.50)*** | 13.28 | (6.85, 25.73)*** |

(*Continued*)

**Table 3.** (Continued)

|  | UOR | 95% CI | AOR[a] | 95% CI |
|---|---|---|---|---|
| No | Ref |  |  |  |

*<0.05

**<0.01

***<0.001

[a]Adjusted for hospitals, age, marital status, education, ethnicity, occupation, religion, family history of cancer, smoking, oral contraceptives, multiple sexual partners, and ever heard of cervical cancer screening.

screen for cervical cancer than unmarried women (AOR = 3.02, CI: 1.12, 8.19, P = 0.03). Women with secondary education were five times more likely (AOR = 5.07, CI:1.82–14.09, P = 0.002) and women with primary education were three times more likely (AOR = 3.02, CI: 1.12, 8.42, P = 0.03) to screen as compared to the illiterate women. Women who have used oral contraceptives were also found to be almost two and a half times (AOR = 2.49, CI: 1.36, 4.39, P = 0.003) more likely to screen for cervical cancer. The women who were aware of cervical cancer screening were 13 times more likely (AOR = 13.28, CI: 6.85, 25.73, P<0.001) to screen as compared to those who had never heard about it.

Table 4 demonstrates the adjusted linear regression model that assessed the factors associated with the knowledge and attitude score among health professionals. The sociodemographic characteristics, work experiences related, and training-related variables were adjusted in the model.

Those who were trained in cervical cancer screening (β = 0.20, CI: 0.44, 2.43, P = 0.005) and those who responded they have outreach activities in their hospital (β = 0.16, CI: 0.22, 2.44, P = 0.015) showed positive association with knowledge scores on cervical cancer.

The female health professionals (β = 0.16, CI: 0.41, 8.16, P = 0.03) and those who responded they have a guideline on cervical cancer screening (β = 0.19 CI: 0.89, 9.53, P = 0.026) were more likely to have a higher attitude for cervical cancer screening. Those who responded they have a treatment facility for cervical cancer (β = -0.17, CI: 7.13, -0.45, P = 0.026), and those who responded they have an outreach activity for cervical cancer screening (β = -0.22, CI: -6.29, -1.62, P = 0.001) had a lower score for attitude toward cervical cancer screening.

## Qualitative results

**Characteristics of the KII participants.** We took 23 KII representing the five hospitals, policymakers, and advocates of cervical cancer screening. The details of the characteristics of the participants are provided in Table 5. The majority of the participants were female and represented tertiary hospitals. Four key informants represented the government and were working in the Family Welfare Division (FWD) and National Health Training Center (NHTC) of the Department of Health Services, MOHP. The senior practicing gynecologists and obstetricians comprised 43.5%, and the senior nurses comprised 30.4% of the total participants.

We used the five major themes based on the logic framework of cervical cancer prevention [35] to provide potential explanation and insights for the quantitative results on service utilization among women, and knowledge and attitude among health professionals toward cervical cancer prevention and screening. The themes included policy, supply, health system, health service providers, and community (Table 6).

Anonymous original quotes from the responses of the participants are included in the description of the findings to provide more context.

**Table 4. Factors associated with knowledge and attitude among health professionals regarding cervical cancer prevention and screening.**

| | Knowledge[a] | | Attitude[a] | |
|---|---|---|---|---|
| | β | 95% CI | β | 95% CI |
| **Age** | -0.26 | (-0.23, 0.003) | -0.01 | (-0.31–0.29) |
| **Gender** | | | | |
| Female | -0.01 | (-1.63, 1.39) | 0.16 | (0.41–8.16)* |
| Male | Ref | | Ref | |
| **Occupation** | | | | |
| Doctor | 0.13 | (-0.32, 2.28) | 0.08 | (-1.90, 4.75) |
| Nurse | Ref | | | |
| **Years of work experience** | 0.20 | (-0.03, 0.23) | 0.109 | (-0.19, 0.46) |
| **Training on cervical cancer screening** | | | | |
| Yes | 0.20 | (0.44, 2.44)** | 0.12 | (-0.29, 4.79) |
| No | Ref | | Ref | |
| **Facility guideline on cervical cancer treatment** | | | | |
| Yes | 0.10 | (-0.45, 2.15) | -0.17 | (-7.10, -0.45)* |
| No | Ref | | Ref | |
| **Facility guideline on cervical cancer screening** | | | | |
| Yes | -0.10 | (-2.77, 0.61) | 0.19 | (0.89, 9.53)* |
| No | Ref | | Ref | |
| **Health education material in the facility** | | | | |
| Yes | 0.05 | (-0.75, 1.67) | -0.01 | (-3.39, 2.79) |
| No | Ref | | Ref | |
| **Community outreach activities in the facility** | | | | |
| Yes | 0.19 | (0.22, 2.05)* | -0.22 | (-6.29, -1.63)** |
| No | Ref | | Ref | |

*<0.05

**<0.01

β = standardized beta coefficient

[a]Adjusted for age, gender, occupation, years of work experience, training on cervical cancer screening, facility guideline on cervical cancer treatment, facility guideline on cervical cancer screening, health education material in the facility, community outreach activities.

**Table 5. Characteristics of the key informants.**

| Characteristics | N = 23 |
|---|---|
| | n |
| **Gender** | |
| Female | 20 |
| Male | 3 |
| **Occupation** | |
| Practicing Obstetricians and gynecologists | 10 |
| Practicing Nurses | 7 |
| Government officials from Department of health services, MOHP | 4 |
| Non-government Public health and cervical cancer experts | 2 |
| **Affiliations** | |
| Gynecological and oncological societies and foundation | 2 |
| Tertiary general hospitals and affiliated medical and nursing schools | 14 |
| Cancer Specialty Hospitals | 3 |
| Department of health services | 4 |

**Table 6. Themes based on the framework for cervical cancer prevention.**

| Themes | Categories | Explanation for quantitative findings |
|---|---|---|
| 1. Policy | 1.1 Implementation gap<br>1.2 Lack of national HPV vaccination program | • Service utilization among women<br>• Knowledge and attitude among health professionals |
| 2. Procurement and supply | 2.1 Tailored implementation<br>2.2 Available resources for trained professionals | • Service utilization among women<br>• Knowledge and attitude among health professionals |
| 3. Health system | 3.1 Delivery<br>3.2 Quality | • Service utilization among women<br>• Knowledge and attitude among health professionals |
| 4. Health care providers | 4.1 Capacity building<br>4.2 Enabling change | • Knowledge and attitude among health professionals |
| 5. Community | 5.1 Community engagement<br>5.2 Behavior change communication | • Utilization of services by women |

## 1. Policy

Almost all participants acknowledged gaps in the existing policies and implementation for cervical cancer prevention services and screening in Nepal.

**1.1 Implementation gap.** Almost everyone admitted that the implementation gaps could have led to low service utilization by women for cervical cancer screening and HPV vaccination in their hospitals and in general.

Some also highlighted the challenges due to the lack of inter-sectoral collaborations leading to implementation gaps for training activities for health professionals which could have affected the knowledge and attitude among the health professionals as identified in quantitative results. Some participants representing private hospitals from this study opined that private academic institutions are often excluded from government programs despite being an integral part of contributing to cervical cancer elimination in the country.

*"There are strategies and policies at the national level. Guidelines and training resources are available. However, private medical colleges are mostly excluded from the national programs on training of health service providers. Many women rely on private teaching hospitals for cervical cancer screening and treatment. We can achieve the targets only if all the sectors are together in the policy implementation."*- Gynecologist, private tertiary teaching hospital

**1.2 Lack of national HPV vaccination program.** Many participants highlighted that delay in national vaccination campaign could be the major reason that had led to extremely low coverage of HPV vaccination in the quantitative results. The policymakers informed that the discussions on introducing HPV vaccination as part of the national program are in process. However, they too acknowledged the delay.

*"In Nepal, prevalence of cervical cancer is increasing because we haven't been able to do as much we can do. Like primary prevention for cervical cancer is vaccination but vaccination program hasn't started from government's side which should have started by now."*- Senior gynecologist, executive, gyne-oncological society

## 2. Procurement and supply

Many participants suggested while the policy is in place, there should also be a balance between procurement and supply for proper implementation to improve service utilization among women and knowledge and attitude among health professionals.

**2.1 Tailored implementation.**    Some suggested that the government should design tailored programs depending on the availability of resources for different health facilities.

*"Policymakers should make policies keeping in mind where the updated technology is available and where it is not. Our hospital has a cancer registry and better technologies and facilities than others. One rule that fits all isn't practical for policy implementation. The services provided by the hospitals with resources and those without are different. Therefore, policy needs to be relevant and contextual to different types of facilities."*-Nurse, government tertiary hospital

The health providers also highlighted the lack of availability of the HPV vaccine despite the increasing demand, especially in private hospitals. They pointed out the imbalance between demand and supply that has led to low HPV vaccination uptake.

*"The HPV vaccine is in demand here. However, we don't have that facility because it's expensive and not affordable for most people. We cannot purchase the vaccines in bulk and store them for a long time due to their expiry dates, but when a few seek vaccine services, we don't have the vaccines to provide them. The government should include HPV vaccination in its national program to address this imbalance."*-Gynecologist, Private hospital

**2.2 Available resources for trained professionals.**    Some also highlighted the need for balance between training coverage and the availability of resources for trained professionals to improve attitude among the health professionals on cervical cancer screening and prevention.

*"We need to understand that training alone is not a miracle. A trained person needs resources and opportunities to utilize their skills. If doctors receive training on colposcopy, then they should have access to a colposcopy machine where they work."*—Senior representative, NHTC

## 3. Health care system

The participants also provided their insights on improving service delivery and quality of care to complement the health providers' knowledge and ways to improve their attitude toward cervical cancer screening.

**3.1 Delivery.**    Many suggested that improving the services in one facility alone might not help in health care system. They highlighted the importance of establishing linkages with different facilities for timely referral.

*"Frequently VIA screening should be done from the hospital and suspected cases to be referred properly. There should not be any complaint that diagnosis was done but we couldn't get treatment. Referral system should improve so that timely diagnosis and treatment are available."*-Gynecologist, Government hospital

**3.2 Quality.**    Many opined that improving the training coverage for health providers could enhance the quality of health services in health facilities. They pointed out that inadequate awareness among health workers remains a challenge which could have affected their knowledge and attitude. They also linked lack of awareness among health professionals to low service utilization by the women seeking health services in the selected hospitals.

*"The health workers have studied about cervical cancer in their curriculum. However, it doesn't mean they remember everything about cervical cancer screening and prevention forever. So we should provide refresher training regularly to raise awareness among health workers. But the training remains inadequate. The government has provided VIA training so some of them could be more aware. But there are many health workers who haven't received any training and are not sensitized about the importance of cervical cancer prevention services and screening and thus are not motivated to provide the service to the women in their clinical settings. More training coverage should be a priority."* - Public health expert, NGO hospital

A few responded that their hospitals are implementing the government policies adequately to improve the quality and are working toward improving service utilization by women.

*"I can say proudly that our hospital is implementing the policies well. As a head of the department of Gynecology, I have ensured VIA and pap smear for every patient who comes to our outpatient clinic. We also organize outreach activities in the community where we have ensured to perform VIA to promote government's policy. If any woman is detected with cancer, we follow the protocols and refer them for treatment as needed."* -Gynecologist, non-government tertiary hospital

## 4. Providers and healthcare workers

Almost everyone expressed that capacity building of the health workers remains a priority and agreed that it is one of the major contributing factors for improving knowledge as identified in the quantitative study. They acknowledged the challenges of inadequate training coverage and suggested alternate training solutions to enable changes.

**4.1 Capacity building.** Many reflected on the improvement in health workers' capacity in recent years after the implementation of national policy. They also opined that many health workers are still not trained due to a lack of trainers to expand capacity building and suggested that it could be the reason behind low training coverage among health providers in the quantitative results of this study affecting their knowledge and attitude.

*"The national policy on cervical cancer prevention services was introduced only in 2010. There were not many trained professionals back then. But with the collaboration with the government, we developed a pool of trainers. We have continued providing training to nurses and doctors. However, the pool of trained professionals remains inadequate to provide screening services to meet the target of reaching at least 50% of women in Nepal. We are aware of this and working on it."* -Gynecologist and national trainer, non-government tertiary hospital

**4.2 Enabling change.** Some also highlighted that the inadequate trained professionals are due to the turnover of trained professionals in many health facilities and the need to repeat training for recruits could have affected the low training coverage as identified in the quantitative results. Some suggested a more efficient way of improving training coverage could be through focused pre-service training when the health professionals are still students which could help improve knowledge and attitude among health service providers when they are ready for the workforce.

*"We can't tell them to work when they don't know how to work. We can promote pre-service training to the nursing and medical students and incorporate the training package into their*

*clinical rotation in OBGYN. If we could do this, we can add mentoring courses as part of in-service training when they go to work. It could fill the gaps created due to the turnover of trained professionals. Otherwise, the gap in inadequately trained professionals would continue."* - Senior representative, NHTC

## 5. Community

Many also acknowledged that improving the services in the health facilities alone is not enough to strengthen the implementation. They highlighted the importance of establishing community linkages through community engagement and behavior change education strategies to improve service utilization among women.

**5.1 Community engagement.**   Almost everyone expressed that strengthening community engagement would enhance awareness among people. The facilities with community outreach programs also reflected that they could provide better services because of outreach community activities as it was identified as a factor influencing the attitude among health providers in the quantitative results.

*"Various outreach awareness activities through virtual or physical presence can improve awareness and demand. I think the outreach activities are helping women utilizing the cervical cancer screening services. However, many women in rural areas are still unaware. Even women living in urban areas like Kathmandu don't prioritize their health. We have been conducting community outreach activities to reach these women and educate them. We are also educating the health workers working in those areas."* –Gyne-oncologist, private hospital

**5.2 Behavior change communication.**   Many participants suggested that more efforts in behavior change communication are also needed to enhance community awareness to improve service utilization. Many opined that public awareness activities are not adequate.

*"Information related to cervical cancer in hoarding board similar to smoking prevention would be effective. If we could create media content related to cancer prevention at a larger scale through advertisements, documentaries, dramas, radio programs, etc., then it could help improving awareness. We can learn the lessons from promoting awareness during COVID time. Similar approaches could raise awareness for cervical cancer prevention services."*- Nurse, Government tertiary hospital

## Discussion

This study showed low utilization of cervical cancer screening and prevention among women attending the outpatient clinics of the selected hospitals. The knowledge among the health professionals was positively associated with their training and community outreach activities related to cervical cancer prevention services and screening in their hospitals. Female health professionals and those having cervical cancer screening guidelines in their hospitals were more likely to have a better attitude toward screening. The qualitative findings complement the quantitative findings and provide further insights into implementation gaps leading to low service utilization among women.

In this study, the utilization of cervical cancer screening through pap smear among women was only 22% which is much lower than the national target. A systematic review and meta-analysis on pooled analysis suggested that 17% of women in hospital-based studies and 16% in

community-based studies utilized cervical cancer screening in Nepal [36]. The factors associated with screening utilization in this study included age, marital status, education, use of oral contraceptives, and having ever heard about cervical cancer screening. The findings in this study are consistent with previous studies conducted in Nepal and globally. A meta-analysis of the global population from 10 cross-sectional studies showed that a higher level of education improves the odds of women utilizing cervical cancer screening [37]. Studies have shown a similar association of cervical cancer screening utilization with socio-demographic status such as age, marital status, and awareness of cervical cancer screening [38, 39].

In this study, 16% of women were aware of the preventive role of HPV vaccination. However, only 1.5% were vaccinated against HPV. As indicated by the qualitative findings, the policy gap due to the lack of a national HPV vaccination program could have contributed to low awareness and HPV vaccine coverage in this study. As of June 2020, 55% of the 194 World Health Organization member countries had already introduced HPV vaccination as part of the national immunization program [40]. However, adequate vaccination coverage is concentrated in high-income countries. Many LMICs like Nepal still haven't introduced the program [40]. As pointed out in the qualitative findings of this study, in Nepal, HPV vaccinations are only available in a few private hospitals where the users need to purchase the vaccine at a higher cost. At the time of data collection in 2022, the key informants from policy levels in this study informed the planning for national vaccination program is in place and had acknowledged the urgency. A nationwide pilot program has been recently introduced in Nepal in September 2023 with the aim of providing vaccination in at least one hospital in each of the seven provinces in Nepal [41]. However, the rolling out of the program for wider coverage could still take time.

The qualitative findings in this study indicated the need for improving hospital-based and community engagement and behavior-change communication to improve service utilization, and knowledge and attitude among health professionals. The interventions on raising awareness in the hospitals and communities have been proven effective in previous multi-center studies focusing on hospital-based post-partum family planning in Nepal [42]. At the hospital level, equal access to information for women across all sociodemographic strata through educational videos in the waiting areas outside the outpatient clinic, and group and one-on-one counselling on the importance of cervical cancer screening could be effective [42]. Lessons can be learned from various behavior change communication interventions for cervical cancer globally to enhance the strategies to raise awareness for cervical cancer in Nepal [43–45].

The training of health professionals in the quantitative study showed a positive association with better knowledge of cervical cancer screening. However, the training coverage was only 37% in this study. The low coverage of formal training activities among health professionals could have led to low screening among women in this study. A study from Somalia showed that only 22.1% of the participants had cervical cancer training during their pre-service education and 16.8% in-service training after graduation [46]. In contrast, the study from Turkey indicated that nearly 75% of the health professionals had training as part of their pre-service education, and almost 67% had in-service training on cervical cancer after their graduation [47]. As suggested by the key informants in the qualitative study more efforts are needed to promote capacity building for health professionals. An education intervention study from Saudi Arabia for nursing students suggested improved knowledge among nursing students regarding cervical cancer screening after the education intervention [48].

The qualitative findings provided further insights into training of health providers and highlighted that incorporating cervical cancer screening in the pre-service curriculum of medical and nursing education and repeated in-service training activities in the hospitals could help improve the overall knowledge of the health professionals. The locally organized, repeated,

hands-on, and simulated training for health service providers in LMICs has been effective in enhancing their skills and confidence with cervical cancer screening, diagnosis, and treatment [49]. Moreover, ongoing sustainable training activities with long-term partnerships of public and private sectors within the health system are necessary to translate health professionals' knowledge into practice [49]. Adapting pre-service and in-service training activities for different maternal and newborn health interventions has proven to be helpful in improving the knowledge and skills of health professionals [50, 51]. Similar, strategies could help enable better implementation of cervical cancer prevention services and screening.

The quantitative findings indicated that the female health professionals and those who responded they have a guideline on cervical cancer screening in their hospitals were more likely to have a better attitude toward cervical cancer screening. Most of the participants in this study were female, and as cervical cancer is a leading cause among women, female professionals could have prioritized the need for screening more as compared to their male counterparts. In contrast, this study also showed those who responded they have a treatment guideline for cervical cancer and community outreach activity for cervical cancer screening in the hospital had a lower attitude toward cervical cancer screening. The hospitals with treatment for cervical cancer could be more focused on the curative aspect rather than the preventive aspects of cervical cancer. Whether the health professionals participating in this study actively participated in the community outreach activities was not explored adequately. The health professionals who were not directly involved in the outreach activities or just involved in treatment may have thought the need to screen has been compensated by the community outreach activities. The qualitative findings in this study have highlighted the importance of community engagement to enhance the implementation. The majority of the interventions for cervical cancer screening in Nepal and globally are community-based [36, 52]. The findings of this study highlight the importance of encouraging health professionals at hospitals to pay equal attention to prevention strategies and also participate in community engagement activities. Moreover, behavioral interventions for health professionals through regular communications and reminders, and training activities could be important steps to be considered by the hospitals to improve the attitude among the health professionals.

## Conclusions

This study showed low utilization of cancer prevention and screening in the five major tertiary hospitals in Kathmandu, Nepal. The health professionals' knowledge showed significant association with training and having guidelines on screening in their hospitals. However, the training coverage remained low. The attitude for screening needed more effort as well. The qualitative results supported the quantitative results and highlighted the existing policies are not being implemented adequately. The findings could pave the way for targeted interventions at tertiary hospital levels.

## Supporting information

**S1 Checklist. COREQ (COnsolidated criteria for REporting Qualitative research) checklist.**
(PDF)

**S1 Table. Knowledge of cervical cancer among health professionals.**
(DOCX)

**S2 Table. Attitude about cervical cancer among health professionals.**
(DOCX)

## Acknowledgments

The authors would like to thank International Federation of Gynecology and Obstetrics (FIGO), Nepal Society of Obstetricians and Gynaecologists (NESOG), and Health Action and Research team for their support. We are also thankful to all the research assistants for their contribution in data collection. We are obliged to all the research participants for their valuable time.

## Author Contributions

**Conceptualization:** Ganesh Dangal, Rolina Dhital.

**Data curation:** Ganesh Dangal, Rolina Dhital, Yam Prasad Dwa, Sandesh Poudel, Jitendra Pariyar, Kirtipal Subedi.

**Formal analysis:** Rolina Dhital.

**Funding acquisition:** Ganesh Dangal, Yam Prasad Dwa, Sandesh Poudel, Jitendra Pariyar, Kirtipal Subedi.

**Investigation:** Ganesh Dangal, Rolina Dhital, Yam Prasad Dwa, Sandesh Poudel, Jitendra Pariyar, Kirtipal Subedi.

**Methodology:** Ganesh Dangal, Rolina Dhital, Yam Prasad Dwa, Sandesh Poudel, Jitendra Pariyar, Kirtipal Subedi.

**Project administration:** Ganesh Dangal, Yam Prasad Dwa, Sandesh Poudel, Jitendra Pariyar, Kirtipal Subedi.

**Resources:** Ganesh Dangal, Rolina Dhital, Jitendra Pariyar, Kirtipal Subedi.

**Software:** Rolina Dhital.

**Supervision:** Ganesh Dangal, Rolina Dhital, Yam Prasad Dwa, Sandesh Poudel, Jitendra Pariyar, Kirtipal Subedi.

**Validation:** Ganesh Dangal, Rolina Dhital, Yam Prasad Dwa, Sandesh Poudel, Jitendra Pariyar, Kirtipal Subedi.

**Visualization:** Ganesh Dangal, Rolina Dhital.

**Writing – original draft:** Ganesh Dangal, Rolina Dhital.

**Writing – review & editing:** Ganesh Dangal, Rolina Dhital, Yam Prasad Dwa, Sandesh Poudel, Jitendra Pariyar, Kirtipal Subedi.

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
