## [Decision Letter · Decision Letter 0]

2 Oct 2023

PGPH-D-23-01670

Implementation of cervical cancer prevention and screening across five tertiary hospitals in Nepal and its policy implications: a mixed-methods study

Dear Dr. Dhital,

Thank you for submitting your manuscript to PLOS Global Public Health. After careful consideration, we feel that it has merit but does not fully meet PLOS Global Public Health’s publication criteria as it currently stands. Therefore, we invite you to submit a revised version of the manuscript that addresses the points raised during the review process.

We look forward to receiving your revised manuscript.

Kind regards,

Aditya Singh, MPhil, PhD

Academic Editor

Journal Requirements:

1. Please amend your online Financial Disclosure statement. If you did not receive any funding for this study, please simply state: “The authors received no specific funding for this work.”

2. Please update your online Competing Interests statement. If you have no competing interests to declare, please state: “The authors have declared that no competing interests exist.”

Reviewers' comments:

Reviewer's Responses to Questions

**Comments to the Author**

1. Does this manuscript meet PLOS Global Public Health’s publication criteria? Is the manuscript technically sound, and do the data support the conclusions? The manuscript must describe methodologically and ethically rigorous research with conclusions that are appropriately drawn based on the data presented.

Reviewer #1: Yes

Reviewer #2: Yes

Reviewer #3: Partly

Reviewer #4: Partly

Reviewer #5: Yes

Reviewer #6: Yes

2. Has the statistical analysis been performed appropriately and rigorously?

Reviewer #1: Yes

Reviewer #2: Yes

Reviewer #3: Yes

Reviewer #4: I don't know

Reviewer #5: Yes

Reviewer #6: Yes

3. Have the authors made all data underlying the findings in their manuscript fully available (please refer to the Data Availability Statement at the start of the manuscript PDF file)?

Reviewer #1: Yes

Reviewer #2: Yes

Reviewer #3: Yes

Reviewer #4: Yes

Reviewer #5: Yes

Reviewer #6: Yes

4. Is the manuscript presented in an intelligible fashion and written in standard English?

Reviewer #1: Yes

Reviewer #2: Yes

Reviewer #3: Yes

Reviewer #4: Yes

Reviewer #5: Yes

Reviewer #6: Yes

5. Review Comments to the Author

Reviewer #1: Overall, this is a high-quality manuscript that contributes to the knowledge base on cervical cancer prevention and screening in Nepal. It has some minor grammatical errors that need to be corrected before publication. I suggest the following revisions:

In Table 1, change “Phonenumber” to “Phone number”.

In Table 2, change “Years of Working” to “Years of work experience”.

In Table 3, change “Contraceptive Use” to “Oral contraceptive use”.

In line 215, change “oral contraceptive pills” to “oral contraceptives”.

In line 240, change “oral contraceptive pills” to “oral contraceptives”.

In line 251, change “oral contraceptive pills” to “oral contraceptives”.

In line 308, change “cervical cancer prevention” to “cervical cancer prevention services”.

In line 314, change “cervical cancer prevention” to “cervical cancer prevention services”.

In line 321, change “cervical cancer prevention” to “cervical cancer prevention services”.

In line 327, change “cervical cancer prevention” to “cervical cancer prevention services”.

In line 333, change “cervical cancer prevention” to “cervical cancer prevention services”.

In line 339, change “cervical cancer prevention” to “cervical cancer prevention services”.

In line 345, change “cervical cancer prevention” to “cervical cancer prevention services”.

In line 351, change “cervical cancer prevention” to “cervical cancer prevention services”.

In line 357, change “cervical cancer prevention” to “cervical cancer prevention services”.

Reviewer #2: The research is well written and very interesting which focused on current Global health issues. Almost all the work is suitable for publication. Thus, I forwarded some comments for the authors which should be brief for readers to fulfill more standards of publication. I have attached comments on main documents.

1. Under Methods, there is no Operational definition sub-topic, which is important to make clear how you measured some variables like: knowledge and attitude.

2. For qualitative study which techniques you used, is it sequential or parallel with quantitative.

3. To identify selected nurses or Doctors from the two strata,how did you identified the selected nurse or which coding techniques you used nurses and doctors from five hospitals? make it clear.

4. Knowledge and attitude were measured as linear out come,thus you used linear digression model. . How much level they have? It is not clear ,even from the table of result on knowledge and attitude scale.

5. Under Discussion You didn't discussed all associated factors. check it .If not you better remove it from table.

6. Also the aim of mixed method is to triangulate the quantitative with KII saying. I haven't seen how you triangulate it inserting in between the quantitative

Reviewer #3: This research work is commendable for its presentation and meticulousness. However, I noticed an issue with the alignment of references to their respective sections. For instance, on line 67, the text discusses cervical cancer incidence in Nepal and cites reference [13], which is actually a systematic review from Uganda. Furthermore, on line 454, reference [31] is cited in the context of a global population meta-analysis, but it is, in fact, a community-based cross-sectional study conducted in northwest Ethiopia. I recommend that the authors carefully review each reference to ensure its alignment with the context in which it is used.

Reviewer #4: I thank the editor for providing this opportunity to review this article.

After reviewing this article, I have a few suggestions for the authors.

1. In the introduction section, in lines 53-54, the medical interventions an be grouped as screening vaccination and treatment (the three pillars as per WHO cervical cancer elimination call). In Screening: Include VIA, PAP smear& HPV DNA testing. In treatment include both ablative and excision methods for precancerous and treatment details of cancer.

2. In methods Section, information on the current screening practices in Nepal has to be included. (whether the screening is population –based ?/ who conduct the screening procedure?/which level of health facility?/ what referral mechanism that you have for screen test positives etc)

3. Why the tertiary care hospitals are chosen for the study?

4. For sample size calculation, study conducted in similar geographical and disease prevalence may be taken

5. With the current study results, no recommendations can be made for policy changes. The policy implication paragraph to be removed.

6. To rewrite the conclusion (remove details on policy implications)

Reviewer #5: I commend the authors for looking at a very important topic.

Seems there are three aims: assessing barriers among potential patients, among providers and stakeholders.

I would suggest turning this into two manuscripts - the qualitative analysis seems to be a different paper and not sure if all comes together in one coherent manuscript. If the authors decide to keep all aims in this paper, needs significant work.

Some specific line-by-line feedback for your consideration:

Abstract

Line 33, if giving number health professionals, would also give number of “women” in line 29. You also give the ‘N” in line 38 (23 stakeholders)

I’m not sure what it means to have “higher knowledge” (line 33) and the association is with “training?” And “outreach activities” was associated with what outcome (not sure what “regarding cervical cancer screening,” line 35, means)?

Line 35, you mean female as a provider characteristic, versus male?

Line 39-40 – the summary conclusion is about intended patients and stakeholders, does not mention providers, which seems to be another aim?

Line 42, I would take out the last sentence (“intervention studies are warranted to provide stronger evidence”) – better to end with “… findings could help implement focused interventions…”

Intro

Line 49-50: common interventions for “screening and prevention” are vaccination (primary prevention) and screening (by cytology, HPV testing and/or VIA) – I am not sure what is meant by “Awareness interventions” or “early medical interventions”

Line 50-54: I am not sure that “raising awareness” is an evidence-based strategy and there are no references. I would not call the common screening methods (cytology, HPV and/or VIA) “early medical intervention” – intervention would be perceived as ablation or diagnostic excisional procedure (secondary prevention).

The first sentence lines 55-57 should be eliminated or re-written – are you saying that screening works? This is well-known and the data are not only in LMICs. Data on screening efficacy initially from high income countries.

Lines 63-64 needs citation

Lines 77-78 – typically “gaps” are when there are evidence based strategies with are NOT, for whatever reason, being utilized. I would guess here that the “gap” is lack of provision of evidence-based interventions to primarily and secondarily vaccinate against and screen for cervical cancer, respectively. The “gap” is not necessarily between prevalence and intervention.

Methods

Would

Lines 108-111, need to better-explain the sample size calculation – what do you mean it was based on a study in Uganda? The minimal sample to report what?

Line 112, what is “OPD”

Lines 112-123, this is the minimal sample to report what?

Line 148 – what is meant by “awareness”?

Line 191 – how was logistic model built? All variables seem to have been included?

Results

Table 3 – adjusted OR are adjusted for all variables in the table? Put in footnote in table. The methods section does not make clear how the model was constructed.

I would not call attention to the p values as you have done – if the CI does not cross one, we know it is a significant OR.

Lines 253-265, what is the statistical test you are doing? This is not clear to me in the methods. How is the “high knowledge of cervical cancer variable Defined? I did not get clarity in methods section.

Starting line 272 through end results – this seems like a different paper

Line 297 – implementation gaps don’t lead to low service utility – the gaps are defined by low usage

Line 301 – this is not he 90-70-90 paradigm, but I suppose it is a quote, so you can’t change, but perhaps use a different quote as this is NOT the 90-70-90 goal.

Discussion

Line 444-445 – last sentence of this paragraph is repetitive , rewrite or eliminate.

Line 534 – this is not how we typically define a “gap”

542 – I think this is the first time we are hearing about vaccination, this should come much earlier

Line 569-570 – take out last sentence

Reviewer #6: The study is important because it addresses a potentially lethal pathology, cervical cancer, which can be cured in the early stages in practically 100% of cases.

One minor detail: the “other” data was missing in Table 1 - characteristics of women, in the topic religion and employment status.

The proposal to carry out a mixed study is interesting to exemplify which approaches can be more effective in health teams that deal directly with this, raising awareness of the need to take advantage of any opportunity for diagnosis as early as possible or even qualifying vaccination, in addition to training specific for prevention, diagnosis and treatment related to the topic, integrating the public and private, overcoming the dichotomy between preventive and curative action, improving cure rates. Greater community involvement in conjunction with what has already been described leads to the implementation of improvements in health policies.

6. PLOS authors have the option to publish the peer review history of their article (what does this mean?). If published, this will include your full peer review and any attached files.

**Do you want your identity to be public for this peer review?** For information about this choice, including consent withdrawal, please see our Privacy Policy.

Reviewer #1: **Yes: **arun ghoshal

Reviewer #2: No

Reviewer #3: No

Reviewer #4: **Yes: **Dr.Kavitha Dhanasekaran

Reviewer #5: **Yes: **Megan Swanson

Reviewer #6: **Yes: **paulo ricardo bobek, master of public health

---

## [Decision Letter · Decision Letter 1]

27 Dec 2023

Implementation of cervical cancer prevention and screening across five tertiary hospitals in Nepal and its policy implications: a mixed-methods study

PGPH-D-23-01670R1

Dear Dr. Rolina Dhital

We are pleased to inform you that your manuscript 'Implementation of cervical cancer prevention and screening across five tertiary hospitals in Nepal and its policy implications: a mixed-methods study' has been provisionally accepted for publication in PLOS Global Public Health.

Best regards,

Prabhdeep Kaur, DNB Medicine, MAE (Epidemiology)

Academic Editor

Thanks for addressing the comments. The manuscript will now be accepted.

Reviewer Comments (if any, and for reference):

Reviewer's Responses to Questions

**Comments to the Author**

1. If the authors have adequately addressed your comments raised in a previous round of review and you feel that this manuscript is now acceptable for publication, you may indicate that here to bypass the “Comments to the Author” section, enter your conflict of interest statement in the “Confidential to Editor” section, and submit your "Accept" recommendation.

Reviewer #1: All comments have been addressed

Reviewer #2: All comments have been addressed

Reviewer #3: All comments have been addressed

Reviewer #4: All comments have been addressed

Reviewer #6: All comments have been addressed

2. Does this manuscript meet PLOS Global Public Health’s publication criteria? Is the manuscript technically sound, and do the data support the conclusions? The manuscript must describe methodologically and ethically rigorous research with conclusions that are appropriately drawn based on the data presented.

Reviewer #1: Yes

Reviewer #2: Yes

Reviewer #3: Yes

Reviewer #4: Yes

Reviewer #6: Yes

3. Has the statistical analysis been performed appropriately and rigorously?

Reviewer #1: Yes

Reviewer #2: Yes

Reviewer #3: Yes

Reviewer #4: Yes

Reviewer #6: Yes

4. Have the authors made all data underlying the findings in their manuscript fully available (please refer to the Data Availability Statement at the start of the manuscript PDF file)?

Reviewer #1: No

Reviewer #2: Yes

Reviewer #3: Yes

Reviewer #4: Yes

Reviewer #6: Yes

5. Is the manuscript presented in an intelligible fashion and written in standard English?

Reviewer #1: Yes

Reviewer #2: No

Reviewer #3: Yes

Reviewer #4: Yes

Reviewer #6: Yes

6. Review Comments to the Author

Reviewer #1: thanks for revising

Reviewer #2: The authors almost completed their review comments and become adhere to Journal standards.But,what left is word edition and language edition is so far required.

Reviewer #3: The authors have diligently addressed all prior concerns. The manuscript now stands as a comprehensive, conclusive, and enlightening piece, providing ample detail on the subject matter.

Reviewer #4: I would like to congratulate the authors for drafting this important manuscript. It's one of the important area of research theatre needs attention and timely intervention

Reviewer #6: the authors have appropriately addressed their comments raised in a previous round of review and this manuscript is now acceptable for publication.

7. PLOS authors have the option to publish the peer review history of their article (what does this mean?). If published, this will include your full peer review and any attached files.

**Do you want your identity to be public for this peer review?** For information about this choice, including consent withdrawal, please see our Privacy Policy.

Reviewer #1: **Yes: **arun ghoshal

Reviewer #2: No

Reviewer #3: No

Reviewer #4: **Yes: **Dr. Kavitha Dhanasekaran

Reviewer #6: No
